# Is Autophagy Involved in the Diverse Effects of Antidepressants?

**DOI:** 10.3390/cells8010044

**Published:** 2019-01-12

**Authors:** Theo Rein

**Affiliations:** Max Planck Institute of Psychiatry, Munich 80804, Germany; theorein@psych.mpg.de; Tel.: +49-89-30622531

**Keywords:** autophagy, antidepressants, depression, cancer, FKBP51

## Abstract

Autophagy has received increased attention as a conserved process governing cellular energy and protein homeostasis that is thus relevant in a range of physiological and pathophysiological conditions. Recently, autophagy has also been linked to depression, mainly through its involvement in the action of antidepressants. Some antidepressant drugs and psychotropic medication have been reported to exert beneficial effects in other diseases, for example, in cancer and neurodegenerative diseases. This review collates the evidence for the hypothesis that autophagy contributes to the effects of antidepressants beyond depression treatment.

## 1. Autophagy

The continuous synthesis and degradation of (macro)molecules to keep internal entropy low is a hallmark of life [1]. While in its first decades, molecular biology focused on the discovery of the fascinating principles of synthetic processes, the equally important mechanisms of degradation are now also the subjects of intense research. Two distinct pathways have evolved for intracellular degradation: the ubiquitin–proteasome system and the autophagy–lysosome system [2]. Over the past couple of years, it has become increasingly evident that the two pathways influence each other [3]. Originally described as a response to calorie restriction [4,5], autophagy is now generally known as a degradative process promoting energy, organelle and protein homeostasis. This essential cellular homeostasis mechanism is described in detail in several excellent reviews [6,7,8,9] and thus will be introduced here only briefly.

The different types of degradative autophagy include macroautophagy, chaperone-mediated autophagy (CMA) and microautophagy. The autophagic machinery is also involved in non-degradative processes, such as the secretion of complex cytoplasmic cargo, referred to as secretory autophagy [10]. Macroautophagy is the most investigated form of autophagy. It is a step-wise mechanism leading to the formation of a double membrane vesicular structure (called autophagosome) that enwraps to be degraded cytosolic material and is then fused with lysosomes to form autolysosomes. This process is enacted by an array of proteins encoded by autophagy related genes (ATGs). The key proteins are the ATG1/ULK complex (ULK1/2 are mammalian homologs of the yeast ATG1 protein), which initiates autophagy by the isolation of membrane material, and Beclin1/ATG6, as part of the nucleation complex that begins the formation of the double membrane phagophore structures. Both ULK and Beclin1 are under the control of the nutrient sensor AMPK (adenosine monophosphate-activated protein kinase), directly and also indirectly through the mTOR complex [11]. The expansion of the phagophore into the autophagosome requires two conjugation systems, ATG7 and ATG10, which catalyze the conjugation of ATG12 with ATG5, and then promote the lipidation of LC3B/ATG8 (conjugation with phosphatidylethanolamine) to form LC3BII. More details of the various steps of the autophagic pathways from membrane isolation to fusion of the autophagosome with the lysosome are provided in various reviews [6,7,8,9,11,12]. Of note, it is increasingly recognized that there is considerable redundancy of autophagic proteins. Under certain circumstances, autophagy can also be observed in the absence of genes originally considered to be essential for the autophagic pathway such as ATG5, ATG7 or Beclin1 [9].

While a broad spectrum of cargo—e.g., damaged proteins, organelles, and pathogens—can be degraded by macroautophagy, CMA is confined to proteins [13]. In essence, CMA degrades proteins with a specific motif at their N-terminus [14]; the chaperone, HSC70, binds to this motif and delivers the protein to the lysosome through interaction with the lysosomal LAMP-2A protein [13,15]. Microautophagy is a much less investigated non-selective lysosomal degradation mechanism that involves enwrapping cytosolic material by the lysosomal membrane and delivery into the lysosomal lumen upon membrane scission [16]. Based on membrane dynamics and the molecular machinery involved, three types of microautophagy are distinguished: microautophagy with lysosomal protrusion, microautophagy with lysosomal invagination and microautophagy with endosomal invagination [17]. There is mechanistic overlap between the different types of degradative autophagy. For example, the binding of HSC70 to the N-terminal amino acid motif of the cargo protein is part of both CMA and endosomal microautophagy [18]. Related to the cross-talk between different forms of autophagy, particular physiological conditions can cause more than one type of autophagy. For example, amino acid depletion has recently been shown to not only induce bulk macroautophagy, but also endosomal microautophagy [19,20].

## 2. Antidepressants

Since autophagy is a fundamental physiological process that is also involved in several pathologies, increasing efforts are being devoted to deciphering pharmacological ways of addressing the autophagic pathway [21,22,23]. Concomitantly, several compounds and drugs were discovered to impact autophagy, including antidepressant drugs [24,25,26,27,28,29]. In general, currently used antidepressant drugs were developed based on the monoamine deficiency hypothesis of depression [30,31]. This hypothesis stipulates a shortage of monoaminergic neurotransmitters in the synaptic cleft. The theory was derived from the observation of the effects of tricyclic antidepressants (TCAs), with imipramine being the first TCA to have been introduced more than 60 years ago [32]. In the following, this hypothesis has stimulated the development of more selectively acting drugs, for example selective serotonin re-uptake inhibitors (SSRIs) [33]. These newly developed drugs exhibit fewer side effects, but no better treatment efficacy [34]. Furthermore, there is some evidence for a sex difference in the effectiveness of these two classes of antidepressants, with men responding better to TCAs than women, while women appear to better respond to SRRIs than men [35].

Efforts to improve drug development based on neurobiological findings have had limited success so far, despite promising developments. For example, one of the most established findings in molecular psychiatry is the association of depression with a dysfunctional hypothalamic–pituitary–adrenocortical (HPA) axis [36,37,38,39]. Therefore, drugs that target components of the HPA axis, such as the glucocorticoid receptor and the receptors for corticotrophin-releasing hormone or arginine vasopressin, have been tested. Overall, controlled large-scale studies are still needed to prove the clinical usefulness of these compounds [40,41,42,43]. It should be noted here that another avenue of HPA axis-derived drug development in depression targets the glucocorticoid receptor inhibitor FK506 binding protein (FKBP) 51 [44,45], which has recently been linked to autophagy [26,27,46]. Furthermore, the N-methyl-D-aspartate receptor (NMDAR) antagonist, ketamine, which has demonstrated efficacy as an antidepressant [47,48,49], has also been linked to NMDAR-independent effects [50], including impacts on autophagy [51,52].

The discovery of the first antidepressants in the 1950s was fortuitous since these compounds were originally developed for a different clinical application [32]. Conversely, following the concept of drug repositioning [53,54], antidepressant drugs are now being explored for their usefulness in diseases beyond depression, for example, in cancer treatment [55]. The following sections survey the reported effects of antidepressants in different clinically relevant conditions with particular consideration of the potential involvement of autophagy.

## 3. Autophagy in the Action of Antidepressants

There are quite a number of publications which report that antidepressants impact autophagy, as has been collated very recently [56]. The majority of these studies report the induction of autophagy, mostly based on the increase in abundance of autophagic markers, typically LC3II/I. However, this conclusion might be premature, as inhibiting functional autophagy by blocking the fusion between autophagosome and lysosome also produces increased levels of this marker. Therefore, the guidelines for the use and interpretation of assays for monitoring autophagy call for the use of methods that determine protein-turnover [57]. The studies employing these methods mostly report increased autophagy upon treatment with antidepressants [26,27,28,29,58], but impaired autophagy has also been reported [25]. Collectively, these studies point to the diverse and cell-type-dependent effects of antidepressants on autophagy.

The molecular mechanism of the autophagy-modulating function of antidepressants is largely unexplored. Nevertheless, an interesting recent study provides evidence for an autophagy-inducing mechanism involving the inhibition of acid sphingomyelinase (ASM) [58], which has long been considered a potential target for antidepressant action [59,60,61,62]. The antidepressants amitriptyline and fluoxetine lead to the gradual accumulation of sphingomyelin in the lysosomes and of ceramide in the endoplasmic reticulum, which stimulates autophagy via protein phosphatase 2A, ULK, Beclin and LC3B [58]. Another study reported the involvement of the stress protein FKBP51 [45,63], Akt1 and Beclin1 in the effect of antidepressants on autophagy and behavior [26,27]. Given the multitude of the cellular effects of antidepressants beyond monoaminergic neurotransmission, and the plethora of at least partially redundant autophagy proteins [9], it will be challenging to pinpoint the exact mechanisms that evoke the autophagic response. Their characteristic as cationic amphiphilic chemicals might play a role [61,64], and new tools to decipher novel molecules directly interacting with antidepressants should also contribute to elucidating their molecular mode of action [64,65].

### 3.1. Depression

In general, it is difficult to unequivocally prove the causal involvement of a particular molecular pathway in the effect of a drug in a clinical setting. Virtually all studies on antidepressants and autophagy in depression have been performed in murine or cellular models, with the known limitations [66] of modeling such a complex disease in animals. The expression of the autophagy initiator and regulator protein Beclin1 [67] in peripheral blood mononuclear cells (PBMCs) of depressed patients was positively correlated with clinical treatment success [26,27]. Moreover, the induction of the autophagic markers Beclin1, pAkt and LC3II/I in response to treatment of these PBMCs with antidepressants was predictive of the clinical treatment response [26,27]. Of the published investigations, this study comes closest to clinical settings, but it cannot prove the causal involvement of autophagy in clinical antidepressant action. It should also be mentioned that the sample size was between 30 and 50, and “only” the antidepressants amitriptyline, paroxetine and fluoxetine were investigated. While the autophagy types were not differentiated, the effects were linked to the Hsp90 co-chaperone FKBP51 [26,27], which is relevant in stress adaptation and depression, as well as in homeostatic pathways in many ways [45,63,68].

In animal studies, it is possible to employ genetic or pharmacological inhibition of autophagy to more rigorously test for a causal contribution of autophagy to the effect of antidepressants in depression-like behavior. In a recent example, deletion of the autophagy regulator ATG5 in microglia did not significantly change the effects of the antidepressant fluoxetine on depression-like effects in a chronic unpredictable mild stress paradigm in mice [69]. On the other hand, the pharmacological inhibition of Beclin1 abrogated the beneficial cellular and behavioral effects of the antidepressants, amitriptyline and fluoxetine, in mice [58]. Further studies testing causalities are awaited. A great number of studies report the effects of antidepressants or antidepressant-like compounds on autophagy, as surveyed very recently [56].

### 3.2. Cell Proliferation, Differentiation and Cancer

#### 3.2.1. Autophagy and Cancer

In general, autophagy is known as a beneficial, cell-protective process [6,7,8,9]. However, excessive autophagy has also been linked to cell death under certain circumstances. This is referred to as type II programmed cell death, which is distinct from apoptosis (type I programmed cell death); several interrelations between apoptosis and autophagy are known [8,70,71]. Furthermore, as discussed elsewhere [22], the observation of massively enhanced autophagy around cell death per se does not prove causality, but often may indicate the final adaptive efforts of the cell. Nevertheless, autophagic cell death has been documented in some cases; for example, deletion of autophagy genes interfered with cell death during Drosophila melanogaster larvae development [72].

Reflecting the potentially dual role of autophagy in cell proliferation and cell death, there is no general rule for the best approach in using autophagy inducers or autophagy inhibitors for cancer treatment [22,73,74]. The development of straightforward concepts may need to take into consideration the stage and the type of tumor, and this is further complicated by the fact that autophagy is intertwined with the immune system that is relevant for the surveillance and suppression of tumor cells [75,76,77,78]. The double-edged function of autophagy is illustrated, for example, by the tissue-specific genetic blockade of autophagy in mice: the progression of lung tumors was markedly impaired, but the onset was accelerated [79]. The promotion of tumor progression by autophagy was also concluded very recently in a murine model of acute myeloid leukemia (AML) where deletion of the autophagy receptor p62 impaired the expansion and colony-forming ability of leukemia cells [80]. Several other preclinical studies support the hypothesis that autophagy inhibitors might be useful in fighting cancer [81,82,83]. Nevertheless, the clinical trials revealed no beneficial effect of the autophagy inhibitors chloroquine and hydroxychloroquine in cancer treatment [22,84].

Conversely, it has been documented in several animal models that autophagy inhibits malignant transformation. For example, blocking autophagy in mice through the systemic mosaic deletion of Atg5 or the deletion of Atg7 in the liver led to the development of multiple liver tumors [85]. Moreover, the heterozygous loss of Beclin1 in mice led to propensity for the development of lung and liver tumors as well as mammary hyperplasia later in life [86,87]. Furthermore, very recent data suggest that annexin A1 suppresses autophagy, thereby promoting migration, invasion and metastasis of nasopharyngeal carcinoma cells [88]. Along these lines, several chemical autophagy inducers are either approved for cancer treatment or in clinical trial [89]. Mimetics of caloric restriction, tapping into prototypic autophagy induction, were shown to enhance the effect of established antitumor drugs and radiotherapy in mice [84,90,91]. Taken together, autophagy appears to show a Janus face in its role in cancer cell survival and death by, on the one hand, inhibiting malignant transformation that may involve its effects on immune surveillance, and on the other hand, accelerating tumor progression in certain cases [22,79,92,93].

#### 3.2.2. Antidepressants and Cancer

Tricyclic antidepressants (TCAs), due to their putative genotoxicity [94] and a few reports of their potential tumor promoting effects in animal models [95,96,97], have raised some concerns in the past about their potential action as carcinogens. These concerns, however, could not be substantiated in several studies. For example, a case-control study using the General Practice Research Database (GPRD) and examining a total of 31,953 cases of cancer (matched to controls) concluded that lung, breast and prostate cancers were largely unaffected, while TCAs may actually display preventive effects for colorectal cancer and glioma [98]. Furthermore, analysis of 7878 cases of epithelial ovarian cancer and 73,913 controls revealed no association between antidepressant use and the risk for this type of cancer [99]. Likewise, antidepressant use was not correlated with a recurrence of breast cancer [100].

Several preclinical studies in cells and animals report anticancer effects of tricyclic antidepressants [24,101,102,103,104,105,106,107]. More recently, a drug repositioning approach revealed the inhibitory effects of TCAs on small cell lung cancer and other neuroendocrine tumors [108]. The involvement of autophagy in the anticancer effects of antidepressants is suggested by several reports. Another and very recent drug repurposing study found that sertraline acts as a sensitizer of non-small cell lung cancer cells to drug (erlotinib) treatment by inducing autophagy [55]. More specifically, treatment of cells with sertraline, erlotinib, or their combination, elevates autophagic flux through reciprocally regulating the AMPK/mTOR pathway and pharmacological or genetic interference with autophagy decreases the anticancer efficacy of sertraline or the combination of sertraline and erlotinib [55]. Similarly, using ATG7 knock down, the tumor growth inhibition by imipramine in combination with another autophagy inducer could be linked to its autophagy promoting effect [109]. Furthermore, sertraline both induces autophagy and inhibits cell growth in the cell lines and primary cells of acute myeloid leukemia, while employing bafilomycin A1 (BafA1) as an inhibitor of autophagy attenuates these effects [110]. In U-87MG human glioma cells, the TCA imipramine exhibited antitumor effects, which were abrogated using siRNA directed against Beclin1 [111]. Likewise, in drug-resistant Burkitt’s lymphoma, the antidepressants maprotiline and fluoxetine induced Type II autophagic cell death, which was diminished by the autophagy inhibitors 3-methyl-adenine (3-MA) or BafA1 [112]. Similarly, fluoxetine induced autophagy and decreased the proliferation and adipogenic differentiation of human adipose-derived stem cells [113]. Autophagy as well as the anti-proliferative effect was attenuated by 3-MA [113]. Fluoxetine was also found to induce autophagic cell death in triple negative breast cancer [114,115], and this effect was also diminished by 3-MA [114].

There are also studies that report the anti-proliferative effects of antidepressants in co-occurrence with autophagy induction, which thus do not attempt to establish causality but suggest a functional link. For example, plant lectins extracted from *Dioclea violacea* display antidepressant-like and antitumor effects and induce autophagy [116]. The antidepressant indatraline was reported to induce autophagy and to inhibit smooth muscle cell proliferation and restenosis in a rat model [117]. In HepG2 cells, amitriptyline induced autophagy, mitophagy and cell death [118]. Finally, inhibition of the autophagic flux by antidepressants has also been reported in conjunction with the inhibition of cell proliferation: the TCA amitriptyline not only inhibits autophagic flux but also endothelial cell proliferation and tube formation [119].

### 3.3. Infectious Diseases, Inflammation and the Immune System

#### 3.3.1. The Role of Autophagy in Infectious Diseases, Inflammation and the Immune System

As alluded to in Section 3.2. on cancer, autophagy exerts non-cell autonomous antitumor effects through the immune system. It enhances the processing and presentation of tumor antigens and also diminishes tumor-promoting inflammation [120]. Autophagy is further established as a defense mechanism in the infection of several bacteria [121,122,123,124]. As an evolutionary response to this host defense, bacteria developed strategies to evade autophagic degradation [121,125,126,127]. Autophagy inducing agents were reported to fight bacterial infection [22]. However, as another evolutionary adaptation, some bacteria enhance autophagy in their hosts for their own metabolic benefit [128,129]. Accordingly, autophagy inhibitors displayed some treatment effects [130]. Likewise, autophagy also plays both pro- and anti-microbial roles in plants [131,132].

Similar observations have been made for viral infections. On the cell-autonomous level, autophagy leads to the efficient degradation of viral particles (also referred to as virophagy) [133,134,135,136,137,138]. Accordingly, some viruses evolved mechanisms to reduce or evade autophagy [22,135,139], and autophagy inducers were shown to fight infection of several viruses [22,134,135]. As with bacteria, some viruses evolved strategies to use autophagy for their own benefit [135,140,141,142]. A more recently described mechanism has been found for enteroviruses that use and partly remodel autophagy for their replication and secretion through secretory autophagy [143,144,145,146,147,148,149]. In line, autophagy suppressing compounds, the natural substance berberine and synthetic derivatives thereof, were reported to exhibit antiviral potency against enteroviruses [150].

#### 3.3.2. Antidepressants and Infectious Diseases, Inflammation and the Immune System

Evidence for a role of inflammatory reactions and the immune system in the pathophysiology of depression has accumulated for several decades [151,152]. For example, psychosocial stress and other adverse events, particularly early in life, which are established risk factors for the development of depression [153,154,155], provoke an immune response involving several molecular pathways in mononuclear lymphocytes and inflammation [156,157]. While the exact inflammatory profile depends on the specific type of trauma [157], elevated inflammation generally correlates with a higher likelihood of developing depression and with its severity [158,159]. Elevated levels of inflammation impact physiology and behavior through several mechanisms involving, for example, synaptic neurotransmission in several brain circuits and the stress hormone axis [160,161].

The inflammatory status has also been linked to treatment response in depression and thus the action of antidepressants. A meta-analysis examining data from 35 studies suggests that increased inflammation contributes to treatment resistance [162]. Very recently, a genetic disposition to both inflammation and antidepressant treatment response was discovered, which differed between the antidepressants escitalopram and nortriptyline [163]. It has been suggested that treatment-resistant depression in particular is linked to elevated inflammation, leading to the proposal to explore the potential of anti-inflammatory treatment in depression [160,161,164,165].

Since antidepressants impact both inflammation and autophagy, and since inflammation is relevant for depression treatment, it appears plausible to hypothesize that antidepressants act through regulating autophagy on inflammation and the immune system. Currently, however, there are very few studies that directly address this question. Using a mouse model of depression, it was reported that several antidepressants not only reduce depression-like behavior, but also inhibit the NLRP3-inflammasome in an autophagy-dependent manner [166]. The deletion of ATG5 in cells abolished the effect of antidepressants on both autophagy and the inflammasome [166]. The anti-inflammatory effects of fluoxetine were also assessed in a brain injury model in rats: fluoxetine induced autophagy as evidenced by the increased levels of Beclin1 and attenuated the activation of the NLRP3 inflammasome [167]. Administration of the autophagy inhibitor 3-MA reversed the beneficial effects of fluoxetine [167].

Linkages between viral diseases and depression have been described on several levels, often suggesting mutual causation [168]. Fluoxetine, which has been reported in numerous studies to induce autophagy, is a potent inhibitor of coxsackievirus replication and other enteroviruses [169,170,171]. However, enteroviruses are reported to make use of autophagy for their secretion, as mentioned above. Thus, the involvement of autophagy in this antiviral effect of fluoxetine is not clear. It is possible that autophagy might be induced to a level that clears out the viral particles before they can exit. Similarly, a potential contribution of autophagy to the other antiviral effects of fluoxetine, such as for hepatitis C and Dengue virus, has not been examined [172,173]

### 3.4. Neurodegeneration

A hallmark of neurodegenerative diseases such as Alzheimer’s, Huntingtin’s and Parkinson’s disease is the accumulation of dysfunctional and aggregation-prone proteins, and it is thus not surprising that autophagy, or rather impairment of autophagy, has been linked to neurodegeneration in numerous studies, as documented in several excellent reviews [22,174,175,176]. Accordingly, the pharmacological induction of autophagy is considered a promising approach and several studies reported beneficial effects in various models [22,174,175,177,178,179].

Antidepressants and other psychotropic medication are frequently prescribed to patients suffering from Alzheimer’s, Huntingtin’s, Parkinson’s and other neurodegenerative diseases [180,181,182]. While these drugs are frequently prescribed to treat co-morbid mood disorders, neuroprotective effects have also been reported, typically in preclinical models. For example, in a mouse model of Huntington’s disease that uses ectopic expression of mutant huntingtin (various models are reviewed in [183]), treatment with a combination of the mood stabilizers, lithium and valproate, produced multiple beneficial effects [184]. In related neuronal and non-neuronal cell models, lithium protected against polyglutamine toxicity [185]. Lithium also mitigated beta-amyloid and associated pathologies in a mouse model of Alzheimer’s disease [186]. This study also found enhanced autophagy upon treatment with lithium [186], like several other researchers in diverse cell and animal models [187,188]. Similarly, in a mouse model of Parkinson’s disease, beneficial effects along with the induction of autophagy were observed upon treatment with lithium in combination with valproate [189]. Of note, these authors also pre-treated the animals with the autophagy inhibitor wortmannin, which dampened the protective effects of lithium and valproate [189]. Similarly, clearance of protease-resistant prion proteins in prion-infected cells by lithium was abolished by the autophagy inhibitor 3-MA [190]. Based on these and other results documenting the neuroprotective and autophagy-inducing actions of lithium, this drug was proposed as a candidate drug for the treatment of neurodegenerative diseases in general [191]. In fact, lithium delayed disease progression in patients suffering from amyotrophic lateral sclerosis (ALS) [192]. The same study also reported neuroprotection in a mouse model of ALS concomitant with autophagy induction [192]. However, as pointed out elsewhere, large controlled trials are required to firmly establish the effect of lithium in neurodegenerative diseases [193]. In addition, there is also evidence for autophagy inhibition by lithium through the inhibition of kinase GSK3β [194]. Thus, it has been reasoned that treatment with lithium should be complemented by the application of rapamycin to ensure autophagy induction [194].

The effects of antidepressants in animal models of neurodegenerative diseases [180,181,182,195,196], for example, the slowdown in disease progression in mouse models of Huntington’s disease [197,198], may also be linked to autophagy. However, in the absence of experimental proof, this remains a hypothesis.

### 3.5. Aging/Longevity

The beneficial effect of autophagy on healthy aging and longevity is well documented in dozens of publications, as surveyed recently [89]. It is very likely that this is not independent of the regulation of disease and metabolic processes, but rather a consequence of it. Similarly, antidepressants may contribute to healthy aging through their effects on various disease conditions. A screen of 88,000 diverse chemicals revealed serotonin receptor antagonists, also used as antidepressants in the clinic, as efficient in extending lifespan in adult *Caenorhabditis elegans* [199]. Even though autophagy has not been investigated in this study, the authors point out that lifespan extension is achieved by blocking certain types of neurotransmission that are implicated in food sensing, thus possibly leading to a state of perceived starvation [199]. As mentioned above, caloric restriction is the prototypic inducer of autophagy restriction [4,5] and has been shown to extend lifespan in several laboratories [200,201,202]. Nevertheless, this does not prove that antidepressants extend lifespan through the induction of autophagy in *Caenorhabditis elegans*. In humans, an analysis of 357 adults with Down syndrome indicated a positive correlation of antidepressant medication with longevity [203]. However, this retrospective study could not establish causality, and autophagy was not assessed.

## 4. Conclusions

Autophagy still enjoys growing attention, not least for its link to a broad spectrum of physiological and pathological conditions. In addition, more and more compounds that modulate autophagy are being revealed, both new molecules and FDA-approved drugs such as antidepressants and other psychotropic drugs. Thus, autophagy induction may be seen as a non-specific side effect of many drugs, including antidepressants. Therefore, it is important to include pharmacological or genetic loss of function approaches to prove the relevance of autophagy in the action of antidepressants or any other drug in future studies. Distinguishing the types of autophagy would also greatly improve our understanding of the role of autophagy in antidepressant action. It should further be noted that not all studies report the induction of autophagy by psychotropic drugs; lithium and tricyclic antidepressants appear to be able to inhibit autophagy in some cases. Thus, wherever possible, turnover-assays need to be performed. Moreover, it appears challenging in several cases to come up with a coherent picture of the physiological events. For example, antidepressants are reported to exert beneficial effects on depression, in part through down-regulating immune function and inflammation. At first glance, this is difficult to reconcile with the beneficial effects in cancer, where intact immune surveillance would be advantageous. In such situations, it is usually argued that the effects are context-dependent. Finally, evidence suggests that autophagy is intimately linked to other cellular processes involving membrane re-organization. Therefore, markers of autophagy might actually be indicators of related mechanisms requiring membrane fusion and scission, such as synaptic neurotransmission. It will be interesting to further advance our understanding of the complexity in this exciting area of research.

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
