# Peer review of "Is Autophagy Involved in the Diverse Effects of Antidepressants?"

_cells, 2019, doi:10.3390/cells8010044_

Round 1

Reviewer 1 Report

The review entitled "Is autophagy involved in the diverse molecular effects of antidepressants? by Theo Rein is a very complete and timely review on the topic. It nicely describes the state of the art. 

I only have a few suggestions:

1. It would be very helpful to describe the pathways of autophagy, even if this has been previously summarized. 

2. Several previous publications linked antidepressants to sphingolipids, which has been also linked to autophagy. This should be presented in more detail. 

Author Response

Response to reviewer 1:

I would like to thank the reviewer for the positive description of the manuscript and the insightful and constructive comments.

Suggestion 1:

“It would be very helpful to describe the pathways of autophagy, even if this has been previously summarized.”

Response:

This suggestion is somewhat similar to one from reviewer #2. There is now a more detailed description of the pathways of macroautophagy (lines 33-49, lines 36-49 are new):

“Macroautophagy is the best investigated form of autophagy which is a step-wise mechanism leading to the formation of a double membrane vesicular structure (called autophagosome) that enwraps to be degraded cytosolic material and is fused with lysosomes to form autolysosomes. This process is enacted by an array of proteins encoded by autophagy related genes (ATGs). Key proteins are the ATG1/ULK complex (ULK1/2 are mammalian homologs of the yeast ATG1 protein) that initiates autophagy by the isolation of membrane material, Beclin1/ATG6 as part of the nucleation complex that begins the formation of double membrane phagophore structures. Both ULK and Beclin1 are under the control of the nutrient sensor AMPK (adenosine monophosphate-activated protein kinase), directly and indirectly through the mTOR complex [11]. The expansion of the phagophore into the autophagosome requires two conjugation systems: ATG7 and ATG10 catalyze the conjugation of ATG12 with ATG5, which then promotes the lipidation of LC3B/ATG8 (conjugation with phosphatidylethanolamine) to form LC3BII. More details of the various steps of the autophagic pathways from membrane isolation to fusion with the lysosome are provided in various reviews [6-9; 11; 12]. Of note, it is increasingly recognized that there is considerable redundancy of autophagic proteins. Under certain circumstances, autophagy can be observed also in the absence of genes originally considered to be essential for the autophagic pathway such as ATG5, ATG7 or Beclin1 [9].”

Suggestion 2:

“Several previous publications linked antidepressants to sphingolipids, which has been also linked to autophagy. This should be presented in more detail.”

Response:

I fully agree with the reviewer. There is now a new section on mechanisms that might link antidepressants to autophagy, which includes a more detailed description of the interesting link to autophagy via sphingolipids (lines 110-116):

“The molecular mechanism of the autophagy-modulating function of antidepressants is largely unexplored. Nevertheless, an interesting recent study provides evidence for an autophagy-inducing mechanism involving the inhibition of acid sphingomyelinase (ASM) [58] which is considered as a potential target for antidepressant action since a long time [59-62]. The antidepressants amitriptyline and fluoxetine thus lead to the gradual accumulation of sphingomyelin in lysosomes and of ceramide in the endoplasmic reticulum which stimulates autophagy via protein phosphatase 2A, ULK, Beclin and LC3B [58].”

Reviewer 2 Report

The study is a comprehensive, well-laid out review of the potential role of autophagy modulation in the biological/therapeutic effects of antidepressant drugs. However, it could benefit from including a short description of the signaling pathways and other molecules involved in autophagy regulation (at least the role of AMPK/mTOR signaling and ATG5, ATG7, and beclin-1, which are mentioned in the manuscript, could be briefly explained). Also, the authors could propose molecular mechanisms possibly responsible for antidepressant-mediated autophagy modulation.  

Author Response

Response to Reviewer #2:

I would like to thank the reviewer for the positive description of the manuscript and the insightful and constructive comments.

Suggestion 1:

The study is a comprehensive, well-laid out review of the potential role of autophagy modulation in the biological/therapeutic effects of antidepressant drugs. However, it could benefit from including a short description of the signaling pathways and other molecules involved in autophagy regulation (at least the role of AMPK/mTOR signaling and ATG5, ATG7, and beclin-1, which are mentioned in the manuscript, could be briefly explained).

Response:

This suggestion is somewhat similar to one from reviewer #1. There is now a more detailed description of the pathways of macroautophagy (lines 33-49, lines 36-49 are new):

“Macroautophagy is the best investigated form of autophagy which is a step-wise mechanism leading to the formation of a double membrane vesicular structure (called autophagosome) that enwraps to be degraded cytosolic material and is fused with lysosomes to form autolysosomes. This process is enacted by an array of proteins encoded by autophagy related genes (ATGs). Key proteins are the ATG1/ULK complex (ULK1/2 are mammalian homologs of the yeast ATG1 protein) that initiates autophagy by the isolation of membrane material, Beclin1/ATG6 as part of the nucleation complex that begins the formation of double membrane phagophore structures. Both ULK and Beclin1 are under the control of the nutrient sensor AMPK (adenosine monophosphate-activated protein kinase), directly and indirectly through the mTOR complex [11]. The expansion of the phagophore into the autophagosome requires two conjugation systems: ATG7 and ATG10 catalyze the conjugation of ATG12 with ATG5, which then promotes the lipidation of LC3B/ATG8 (conjugation with phosphatidylethanolamine) to form LC3BII. More details of the various steps of the autophagic pathways from membrane isolation to fusion with the lysosome are provided in various reviews [6-9; 11; 12]. Of note, it is increasingly recognized that there is considerable redundancy of autophagic proteins. Under certain circumstances, autophagy can be observed also in the absence of genes originally considered to be essential for the autophagic pathway such as ATG5, ATG7 or Beclin1 [9].”

Suggestion 2:

“Also, the authors could propose molecular mechanisms possibly responsible for antidepressant-mediated autophagy modulation.”

Response:

I fully agree with the reviewer, and this suggestion also overlaps somewhat with a comment from reviewer #1. There is now a new section on possible mechanisms that might link antidepressants to autophagy (lines 110-123):

“The molecular mechanism of the autophagy-modulating function of antidepressants is largely unexplored. Nevertheless, an interesting recent study provides evidence for an autophagy-inducing mechanism involving the inhibition of acid sphingomyelinase (ASM) [58] which is considered as a potential target for antidepressant action since a long time [59-62]. The antidepressants amitriptyline and fluoxetine thus lead to the gradual accumulation of sphingomyelin in lysosomes and of ceramide in the endoplasmic reticulum which stimulates autophagy via protein phosphatase 2A, ULK, Beclin and LC3B [58]. Another study reported the involvement of the stress protein FKBP51[45; 63], Akt1 and Beclin1 in the effect of antidepressants on autophagy and behavior [26; 27]. Given the multitude of cellular effects of antidepressants beyond monoaminergic neurotransmission, and the plethora of at least partially redundant autophagy proteins [9], it will be challenging to pinpoint the exact mechanisms that evoke the autophagic response. Their characteristic as cationic amphiphilic chemicals might play a role [61; 64], and new tools to decipher direct interaction molecules should also contribute to elucidating their molecular mode of action [64; 65].”